# Fyn Tyrosine Kinase as Harmonizing Factor in Neuronal Functions and Dysfunctions

**DOI:** 10.3390/ijms21124444

**Published:** 2020-06-22

**Authors:** Carmela Matrone, Federica Petrillo, Rosarita Nasso, Gabriella Ferretti

**Affiliations:** 1Division of Pharmacology, Department of Neuroscience, School of Medicine, University of Naples Federico II, 80131 Naples, Italy; rosarita.nasso@unina.it (R.N.); gabriella.ferretti@unina.it (G.F.); 2Department of Biomedicine, Aarhus University, 8000 Aarhus C, Denmark; au621506@uni.au.dk

**Keywords:** Fyn tyrosine kinase, plexins, semaphorins, synaptic plasticity, neurodegeneration, Alzheimer’s disease

## Abstract

Fyn is a non-receptor or cytoplasmatic tyrosine kinase (TK) belonging to the Src family kinases (SFKs) involved in multiple transduction pathways in the central nervous system (CNS) including synaptic transmission, myelination, axon guidance, and oligodendrocyte formation. Almost one hundred years after the original description of Fyn, this protein continues to attract extreme interest because of its multiplicity of actions in the molecular signaling pathways underlying neurodevelopmental as well as neuropathologic events. This review highlights and summarizes the most relevant recent findings pertinent to the role that Fyn exerts in the brain, emphasizing aspects related to neurodevelopment and synaptic plasticity. Fyn is a common factor in healthy and diseased brains that targets different proteins and shapes different transduction signals according to the neurological conditions. We will primarily focus on Fyn-mediated signaling pathways involved in neuronal differentiation and plasticity that have been subjected to considerable attention lately, opening the fascinating scenario to target Fyn TK for the development of potential therapeutic interventions for the treatment of CNS injuries and certain neurodegenerative disorders like Alzheimer’s disease.

## 1. Introduction

Fyn is a non-receptor or cytoplasmatic tyrosine kinase belonging to the Src family kinases (SFKs) consisting of 11 members (Blk, Brk, Fgr, Frk, Hck, Lck, Lyn, c-Src, Srm, and c-Yes) in humans [1].

Src oncogene has been identified and characterized in the early 20th century when Peyton Rous discovered a particle smaller than a bacterium, which later became known as the Rous chicken sarcoma virus, that could be transmitted from bird to bird. Rous’ virus (RSV) was later found expressed in a truncated evolutionary conserved form called c-Src (Src) in uninfected vertebrate cells [2]. Indeed, Src is highly conserved among metazoans, and a Src ortholog has also been shown expressed in unicellular choanoflagellates [3,4].

In mammals, Src, Fyn, and Yes (SYF) are ubiquitously expressed, while the other family members display more restricted expression profile [5].

Fyn is primarily involved in several transduction pathways in the central nervous system (CNS) including myelination, axon guidance, and oligodendrocytes formation. Indeed, when disrupted these pathways, Fyn can contribute to the development of severe brain pathologies, such as Alzheimer’s disease (AD) and multiple sclerosis (MS) [6,7,8].

In the peripheral immune system, Fyn plays an important role in the regulation and functions of T-cell and B-cell receptor signaling and in the differentiation of natural killer cells [1,9,10]. A number of additional biological functions in which Fyn activity is involved has been extensively reported and includes growth factor and cytokine receptor signaling, ion channel function, platelet activation, fertilization, entry into mitosis [11,12].

On the other side, Fyn upregulation and genetic alterations have been associated with some malignancies as well as to several neuronal dysfunctions. In fact, when overexpressed, Fyn influences cell growth and proliferation and causes morphogenic transformations and alterations of mitogenic signals [13,14,15,16]. In addition, Fyn controls integrin adhesion and cell-cell interactions, all clinical features identified in cancer [15,17].

In this review, we will focus our interest on the role exerted by Fyn in the brain. The complexity of Fyn in the brain is mirrored in a myriad of neurological activities. However, we, by no means, expect to guide the reader through all of them, but rather highlight the role of Fyn either in synchronizing and optimizing functional neuronal networks or in exacerbating impaired or dysfunctional pathways in neuronal diseases that have no current cure. To this aim, several studies on the function that Fyn plays in developing neurons will be explored and emerging evidence about the role of Fyn in mature neurons as well as in the onset of brain disorders will be reviewed and discussed.

## 2. Fyn Structure and Activation Mechanisms

Fyn is 59kDa non-receptor protein tyrosine-kinase (TK) member of Src family [15] comprising 537 amino acids encoded by the *Fyn* gene, located on chromosome 6q21 [18]. Three isoforms of *Fyn* have been shown to arise via alternative splicing of exon 7 [15]: the isoform 1 (*Fyn*[B], “canonical sequence”) is the first identified; the isoform 2 (*Fyn*[T]) that tends to be expressed in T-cells and differs from the isoform 1 in the linker region between the SH2 and the SH1 domain [11]; the isoform 3 has been found in the blood cells and differs from the isoform type 1 as missing the sequence 233-287 [18]. *FynT* contains exon 7B (159 bp) and is expressed in thymocytes, splenocytes, and some hematolymphoid cell lines, while *FynB* (168 bp) includes exon 7A and it is more ubiquitous in its expression, although it accumulates principally in the brain [15].

Fyn, similarly, to the other Src members, is composed of several functional parts connected in a single protein chain (Figure 1). Beside the catalytic domain (SH1), Fyn contains two small, mutually unrelated, non-catalytic domains called SH2 and SH3 (Src-homology regions 2 and 3), of about 100 and 60 amino acids, respectively. SH2 and SH3 domains interact with other proteins, and these interactions regulate the TK activity.

Fyn exists in two conformations: active and inactive. Fyn activation (as for the other Src members) depends on the ligand binding to the SH2 and/or SH3 domains and on the phosphorylation/dephosphorylation of two critical tyrosine residues, Tyr528 in the isoform 2, FynT, (corresponding to Tyr531 in isoform 1, FynB) and Tyr417 (the activating Tyr residue) of the isoform 2, FynT, (corresponding to Tyr420 in isoform 1, FynB).

Multiple regulator mechanisms have been proposed leading to Fyn activation. Indeed, the dynamic control of Tyr420 and Tyr531 phosphorylation in the brain provides an important level of regulation of Fyn activity and its ability to interact with other proteins. When Tyr531 is not phosphorylated, Fyn adopts an “open” conformation that allows the interaction of the catalytic domain and the SH2 domain with their specific substrates (Figure 1). Conversely, when Tyr531 is phosphorylated, Fyn folds such that the tail and the SH2 domain interact to each other, preventing the two domains to bind their substrates and making Fyn inactive [9,19,20]. This suggests that increasing the local concentration of either an SH2 or an SH3 ligand would be expected to shift the equilibrium toward the “on” state. Particularly, potent activators would have binding sites for both the SH2 and SH3 domains, and consequently, decreased levels of such ligands would allow Fyn to revert to “the off state” [8]. This destabilization of SH2-SH3 interaction has been suggested as one of the most important mechanisms of Fyn activity regulation [11].

It is not worth that, the replacement of Tyr531 with phenylalanine is sufficient to convert Fyn protooncogene to oncogene [35,36].

Csk is a 50 kDa cytosolic TK differing from the other Src members because of the lack in both, the myristylation and the unique sequences [37,38,39]. Csk phosphorylates Fyn at Tyr531 [40] and promotes the intermolecular binding between SH2 domain and the C-tail, ultimately forcing the kinase into the inactive state [41]. Differently, the striatal enriched phosphatase (STEP), dephosphorylates Fyn at Tyr420 residue in the postsynaptic densities (PSDs) of striatal neurons and thus inactivates Fyn [42].

On the other side, Platelet-Derived Growth factor (PDGF) has been reported to stimulate the intrinsic kinase activity of Fyn by promoting its Tyr420 phosphorylation at *N*-terminal tail thereby activating Fyn intrinsic kinase activity and allowing it to interact with other proteins [16,43]. In addition, PDGF promotes Fyn-dependent migration of oligodendrocyte (OL) progenitors either during the brain development or lesion repair, when OLs migrate long distances before reaching the site of myelin formation [44].

Of note, an intriguing role of PDGF in initiating Fyn DNA synthesis has been also reported [45].

In addition, PDGF-mediated Fyn phosphorylation activates the serine/threonine Cdk5, which phosphorylates microtubule-associated proteins (MAP) such as MAP1B and promotes both microtubule assembly and stabilization during the early phases of neurons and OL differentiation and migration [43,46].

Interestingly, the receptor-like protein tyrosine phosphatase α (PTPα), highly expressed in the brain, regulates Fyn activity by triggering the dephosphorylation of Tyr531 residue on the Fyn C-tail [47,48,49]. PTPα has a very short extracellular domain with no adhesion motifs, which is the reason why it differs from most of the other receptor-like PTPs [48,50,51,52]. Overexpression of PTPα results in the acquisition of oncogenic Fyn phenotype [47] and PTPα deficient mice show a reduction in Fyn kinase activity [53]. PTPα mutant mice, in which the cysteine residues 414S (C414S) and 704S (C704S) are replaced with serine in the catalytic domains active site, fails to dephosphorylate Fyn [49,53].

Notably, Wang and colleagues recently described a novel critical upstream regulator of Fyn signaling, Receptor-type protein tyrosine phosphatase alpha (RPTPα), that appears to be required for oligodendrocyte progenitor differentiation and myelination [54].

## 3. Fyn in the Brain: Distribution and Function

Fyn is one of the highest expressed Src TK in the brain with a distribution in all the limbic regions as well as in cerebellum and striatum [55,56].

During embryonic development, a large amount of Fyn has been detected in the cerebral cortex, in the cerebellum, telencephalon, and brain stem [55,56,57]. Such extensive Fyn distribution is evident during brain development and it persists in adult brain (Figure 2) [55].

The broad distribution of Fyn in the brain reflects its critical role. Indeed, Fyn tempers excitatory and inhibitory synaptic transmission stimuli and regulates mechanisms related to learning and memory processes [55,58,59] (Figure 3).

Consistently, Fyn-deficient mice, show an aberrant distribution of neocortical neurons at E16 specifically in layer II-III [60] with a thin cerebral cortex, axonal degeneration, and a decreased number of oligodendrocytes [61]. Fyn mutant mice carrying a mutation on the regulatory residue Tyr531 (Y531F) display early lethality, reduced weight, hyperactivity, persistent tremor, lack of coordination and altered locomotion behavior [56]. In addition, mice carrying a point mutation in the SH2 domain (FynR176A) show an impaired neural migration in the cerebral cortex [62].

### 3.1. Fyn Promotes Myelination in the CNS

A large body of evidence denotes a critical role of Fyn in myelination [34,63,64,65,66].

Myelin consists of multiple concentric layers of glial cell membrane, highly enriched in lipid (about 70% of the dry weight of myelin) and containing approximately 30% of proteins [67].

Myelin basic protein (MBP) [68] and the proteolipid protein (PLP/DM20) [69], are the two major myelin proteins in the CNS. MBP represents the 30% of total protein content and it is localized at opposing cytoplasmic faces of the myelin lamellae thus playing a role in myelin compaction and in the adhesion of the internal leaflets of the specialized oligodendroglia plasma membrane [70,71,72,73,74].

Umemori et al. (1999) [75] and White et al., (2008) [64] described the involvement of Fyn in the transactivation of the MBP gene in the initial stages of myelination. Shortly, MBP mRNA translation is essential for myelination and this process is regulated by the binding of the trans-acting factor heterogeneous nuclear ribonucleoprotein (hnRNP) A2 to the cis-acting A2 response element (A2RE). Hence, Fyn phosphorylates the hnRNP-A2, thus stimulating the MBP translation and initiating myelination processes [64].

In support to the critical role exerted by Fyn in myelination, Fyn deficient mice show a severe hypomyelination in the forebrain and a significant reduction in the amount of MBP compared to wild type [76]. Conversely, Fyn overexpression triggers myelinization in oligodendrocytes [76].

Interestingly, Lyn and Src knockout (KO) mice show no significant deficit in myelin formation, featuring the possibility that Fyn is the only Src family member that plays a role in myelination [66].

In addition, Fyn kinase-dead mutant mice, with a point mutation in the Fyn ATP binding site (FynK296R), show a severe myelin deficit in the forebrain with a reduced number of oligodendrocytes [66], suggesting that Fyn is required for the oligodendrocyte maturation. In agreement, Osterhout et al., (1999) revealed that Fyn activation is one of the earliest events as oligodendrocyte progenitor cells differentiate [77].

Fyn activation may occur by several pathways. Particularly, mechanisms involving neuronal adhesion molecule L1 [64], integrins [78,79] or gamma chain of immunoglobulin receptor (γFcR) [80] have been previously reported. In particular, the evidence that γFcR activates Fyn signaling cascade and consequently promotes OL differentiation and myelin repair has opened an unexplored scenario for the use of γFcR in the treatment of demyelinating diseases [80].

Of interest, a regulatory mechanism of MBP expression via Fyn mRNA transcriptional inhibition has been described. Accordingly, MiR-125a-3p, that it is particularly abundant in the CNS, directly binds the 3′UTR region of Fyn mRNA and inhibits Fyn expression [81]. This, in turn, controls MBP transactivation, resulting in a delay in the process of oligodendroglia maturation [81].

Another myelin constituent is MAG (myelin-associated protein), an adhesion member of the immunoglobulin superfamily expressed exclusively in myelinating oligodendrocytes and Schwann cells [82]. MAG binds Fyn at SH2 and SH3 domains and triggers Fyn activation [63]. Interestingly, an impressive colocalization of Fyn and MAG in fiber tract and oligodendrocytes during the early stage of myelination has been reported [63]. MAG appears to coordinate OL survival and differentiation through Fyn signals during the initial phase of myelination. Consistently, the genetic abrogation of MAG and Fyn causes a diffuse hypomyelination and exacerbates the phenotype observed in Fyn KO mice [64].

Whether or not Fyn contributes to the severity of diseases in which myelin growth or structure is damaged, such as multiple sclerosis (MS) or ischemic and traumatic brain injury [83], or neuropsychiatric diseases such as schizophrenia [84] or neurodegenerative diseases, such as Alzheimer’s disease [85,86] is largely controverted. Of interest, some single nucleotide polymorphisms (SNPs) in Fyn gene and/or in Fyn-related genes have been detected in schizophrenic patients pointing on Fyn genetic variants as a potential susceptibility factor in myelination disorders [87,88,89].

### 3.2. Fyn Mediates Oligodendrocytes Differentiation and Maturation

Fyn is involved in differentiation and/or maturation of oligodendrocytes (OL) [77]. Studies on primary cultures of differentiating oligodendrocytes demonstrate that Fyn is expressed in the cell body and throughout the processes, both in progenitor and mature oligodendrocytes. In particular, Fyn expression is two or three-fold more, and its kinase activity is 10-30 times higher in mature OL rather than in progenitors [77]. Accordingly, Fyn inhibitors, such as pyrazolopyrimidine derivates PP1 and PP2, prevent OL differentiation causing thick and irregular myelinization and the formation of clamps [77].

In addition, also the number of OL is regulated by Fyn as immunostaining experiments reveal that the absence of Fyn results in a reduction of 40–50% of the number of OL. However, whether this reduction is due to a reduced proliferation or to an increased neuronal death, or even to other mechanisms deserve further clarifications [61,66].

Relevantly, Klein et al. (2002) [90] reported that Tau binds the Fyn SH3 domain, and both are expressed in OL processes and soma. The deletion of -PPXX- Tau motif results in the disruption of Fyn-Tau interaction and causes the decrease in the number and length of OL processes thus emphasizing the significance of Fyn-Tau interaction also in OL process formation [90].

Indeed, alterations in Fyn-Tau interaction have been associated with the onset of some neurodegenerative diseases, including MS [91]. In addition, GWAS datasets have implicated SNPs along Fyn gene as MS susceptibility factors [92,93].

### 3.3. Fyn and Semaphorins in Neurodevelopment and Neurodevelopment-Associated Disorders

Fyn has been largely associated with neurodevelopment and in particular alterations in its expression levels of activity have been linked to the onset of deficits in the behavioral and emotional spectrum. Indeed, Fyn deficient mice have shown abnormal hippocampal development [94] and mice in which the Fyn gene is replaced by LacZ, show an increased sensitivity to stress and emotional abnormalities, also justified by the presence of lesions in the limbic system [56,57,60,95]. Consistently, increased expression of Fyn in the prefrontal cortex seems to contribute to the pathogenesis of schizophrenia [96] and SNPs along the Fyn gene sequence have been related to several development-related disorders [97]. In this context, rs6916861, rs3730353, and rs706895 Fyn polymorphisms were found risk factors for schizophrenia in the Chinese-Han population [98] and rs6916861 and rs3730353 for bipolar disorder [88].

Fyn controls also the BDNF/TrkB and NGF/TrkA pathways which are both involved in neurodevelopment as well as neurodegenerative processes [99,100,101,102,103]. Of note, BDNF gene polymorphisms have been hypothesized to play a possible role in the pathogenesis of autism spectrum disorder (ASD) via Fyn activity dysregulation [104,105,106,107,108,109,110].

Interestingly, Fyn promotes semaphorin receptor (plexin) phosphorylation and facilitates semaphorin and plexin interaction [111,112,113] that is crucial in mediating events related to neuronal polarization and migration [114,115], synapse formation [116,117], axonal pruning [118,119] and dendritic arborization [120,121,122] (Figure 4).

By contrast, in pathological conditions, following mechanisms still not well characterized, Fyn mediates Plexin-A2 hyperphosphorylation thus activating downstream signaling that have been associated with several neurodegenerative diseases such as Alzheimer’s and Parkinson’s disease, multiple sclerosis and Amyotrophic lateral sclerosis [123,124,125,126]. Adapted from Sasaki et al. (2002) [111].

Of relevance, Schafer et al., identified genetic variants of Fyn, as well as semaphorin and plexin, in patients with neurodevelopment-associated disorders and autism [97,113,127,128] (Table 1).

Fyn after interacting with plexin tyrosine sites, activates cyclin-dependent kinase 5 (Cdk5) [111]. Cdk5 is a member of proline-directed serine/threonine kinase family, expressed in proliferating cells that, unlike other family members, has a restricted role in the nervous system [129]. In turn, Cdk5 phosphorylates Tau protein and activates a cascade of events finally resulting in growth cone collapse. Of note, Fyn or Cdk5 pharmacological inhibitors prevent all these events [111] (Figure 4).

On the other hand, Cdk5 also phosphorylates other proteins, such as collapsin response mediator protein-2 (CRMP-2) [130] that appears to be important in mediating Sema3A signaling and in the control of axon formation and microtubule polymerization through its binding to tubulin heterodimers [131,132] (Figure 4). A mutation in Ser522 prevents Cdk5 phosphorylation and protects from Sema3A-induced growth cone collapse [133].

Emerging evidence suggests a role of Sema3A in promoting dendritic branching and neurites formation in cortical neurons via Fyn activation both these events are not observed in neurons exposed to PP2 (Fyn selective inhibitor) or in Fyn^‒/‒^ mice [120]. Of note, studies on Fyn^‒/‒^ or Sema3A^‒/‒^ mice, show that the lack of these two proteins decreases dendritic spine density in layer V of pyramidal neurons, whereas layer III seems not to be affected, highlighting the importance of Sema3A-Fyn pathway in regulating spine morphogenesis and density in cerebral cortex [120].

Although there are, as yet, no examples of neuronal diseases in which alterations in Fyn signal affects semaphorin-plexin interactions and downstream pathways, these are likely to exist.

Dysregulation of semaphorins has been detected in damaged CNS axons and particularly their expression has been found strongly upregulated in MS oligodendrocytes located near the injury site [134,135] (Figure 4).

Changes in the endothelial tight junction structure on the blood–brain barrier (BBB) have been detected in MS brains and associated with Sema3A and Sema4D overactivation [134,135]. Of note these changes have been related to the aberrant activation of plexin-Fyn-AKT signaling pathway and TK inhibitors have been proposed as potential strategy to ameliorate these deficits [7] (Figure 4).

An abnormal neurohistological pattern of semaphorins has been detected in adult brain tissues and in the affected cortex and hippocampus of AD brains [136]. In particular, Sema3A was found hyperphosphorylated in AD brains along with plexins, CRMP-2, and microtubule-associated protein 1B (MAP-1B), that are all molecular targets of Fyn [137]. Of note, semaphorin (as well as their receptor) gene variants have been associated with AD [138], opening the relatively underexplored possibility that alterations in semaphorin pathways may become potential susceptibility factors of AD.

In addition, Venkova et al. hypothesized a role of Sema3A in promoting distal axonopathy and muscle denervation observed in the SOD1G93A mouse model of ALS [139]. This was described to occur through the activation of Fyn/Cdk5 and Glycogen synthase kinase 3 beta (GSK3β) pathways as well as CRMP-2 phosphorylation ultimately leading to microtubule instability and actin cytoskeletal rearrangements [42] (Figure 4).

### 3.4. Fyn Controls Neuronal Migration

Fyn plays an important role in neuronal migration and cortical lamination [62].

A point mutation in the Fyn SH2 domain (FynR176A) impaired neuronal migration and neuronal morphogenesis by inducing neuronal aggregation and branching [62]. In addition, the double KO mouse of Fyn, as well as of Src, shows a reeler-like phenotype [140].

Reelin is an extracellular glycoprotein with a prominent activity in the control of neuronal migration and cellular layer formation in the developing brain [141,142,143]. Reeler mutant mice lacking reelin expression [144] exhibit a neurological phenotype characterized by ataxia and a typical “reeling” gate consisting in widespread defects in neuronal lamination in the developing forebrain, and in cerebellar hypoplasia, due to the failure of radially-migrating neurons to reach their destination and to the failure of Purkinje cells to form a cellular layer, respectively [145,146]. Similar phenotypes have been described in patients carrying reelin homozygous mutations, characterized by lissencephaly with cerebellar hypoplasia [147].

In addition, reelin activates RasGRF1/CaMKII pathway [148] and promotes dendrites maturation, synaptogenesis, synaptic transmission, and plasticity, thus controlling the formation and function of synaptic circuits during the development and in adult brain [149,150,151,152]. Consistently, heterozygous reelin mutations have been associated with lateral temporal epilepsy [153], and have been pointed as a risk factor in autism [154]. The molecular composition of the dendritic spines is also affected by reelin via an unidentified mechanism involving NMDA receptor [155,156].

A crosstalk between Fyn and reelin has been well characterized and elucidated and it is largely dependent on Disabled-1 (Dab1), whose phosphorylation is required for neuronal migration. Phospho-mutant Dab1 mice [157,158], double Fyn/Src KO mice [140], as well as spontaneous or genetically engineered Dab1 KO mice, all show similar reeler-like phenotypes [159,160,161,162,163,164].

Dab-1 binds the cytoplasmic tail of lipoprotein receptors, including ApoER2 and VLDLR [144] and upon reelin binding, becomes phosphorylated on tyrosine residues by Fyn and Src [157].

The binding of Reelin to ApoER2 triggers ApoER2, Dab1, and NMDA receptors clustering and the activation of Fyn, finally resulting in the NMDAR phosphorylation, in an increased Ca^2+^ influx and in neurotoxicity [165,166].

Other molecules have been implicated in reelin-dependent dendrite outgrowth such as the amyloid precursor protein (APP) [167], which binds Dab1 via its cytoplasmic tail [157,168]. Reelin signaling also exerts a protective effect against β-amyloid at the synapse, underscoring the potential relevance of this “developmental” factor for neurodegenerative disorders [149,169,170,171].

### 3.5. Fyn in Synaptic Regulation and Dysregulation

Strategically located at the postsynaptic density of glutamatergic synapses, Fyn is conceived to target local synaptic substrates and to regulate the strength and efficacy of synaptic transmission [172]. Fyn phosphorylates ionotropic glutamate receptors and other synaptic proteins, thereby modulating their expression and synaptic signaling [8,173]. Consequently, deregulations or dysfunctions in Fyn signaling have been related to synaptic deficits or neurodegenerative processes [6,174,175,176,177].

There are three classes of iGluRs: α-amino-3-hydroxy-5-methylisoxazole-4-propionic acid receptors (AMPAR), *N*-methyl-d-aspartate receptors (NMDAR), and kainate receptors [178].

NMDARs form functional channels gathering GluN1 (formerly known as NR1) with GluN2 subunits, mainly GluN2A (NR2A) and GluN2B (NR2B). AMPARs are assembled by four subunits (GluA1–4, previously named GluR1–4) [178,179].

All NMDAR and AMPAR subunits display an intracellular domain that has been found to accommodate dynamic protein–protein interactions and tyrosine phosphorylation [173,180,181,182].

In particular, Fyn phosphorylates the NR2A and NR2B subunits and consequently controls the subcellular or subsynaptic distributions of NMDA receptor and potentiate NMDAR currents [183]. Indeed, among 25 tyrosine residues in the C-terminal cytoplasmic region of NR2B, 7 of these are phosphorylated by Fyn in vitro [184].

NMDA phosphorylation is promoted by the presence of two PSD proteins: PSD-93 and PSD-95 [185,186]. PSD-95 forms a complex with Fyn and NMDA that appears to enhance NMDA activity [187]. Of note, a genetic deletion of PSD-93, results in a lower expression of Fyn and in the reduction in NR2A and NR2B phosphorylation [185].

After ischemic insults in the adult brain, Fyn promotes assembly and remodeling of the PSD complexes [19]. Fyn interacts directly with the α1c subunit of L-type voltage-gated calcium channel (L-VGCC) and increases Ca^2+^ influx and cytotoxicity [188]. Fyn also interacts with PSD-95 associated GTPase, SynGAP that has been reported increased during ischemic events, further triggering neurotoxic processes [189]. In addition, PSD-93 and Fyn/NR2B association and NR2B tyrosine phosphorylation is increased in adult brain after ischemia [190].

As a consequence, tyrosine-phosphorylated PSD-93 binds to Csk, a negative regulator of Fyn activity, and thereby inactivates Fyn [191].

Fyn phosphorylation at Tyr1472 (Y1472) of NR2B subunit, is also important to stabilize synaptic localization of NMDA receptor, by preventing the interaction with clathrin adaptor protein (AP2) and the consequent internalization [192,193,194,195].

Consistently, NR2B phosphorylation at Tyr1472 and Tyr1336 (Y1336) sites selectively enriched NR2B/NMDAR abundance in synaptic versus extrasynaptic compartments, respectively [195].

Of relevance, the aberrant Fyn-mediated phosphorylation of NR2A at Tyr1252 (Y1252) and NR2B at Tyr1472, as well as increased calpain activity, are associated with brain injury and mortality in response to neonatal hypoxic ischemia (HI) [19,196].

In particular, phosphorylation of Tyr1472 contributes to excitotoxic cell death by increasing NMDAR responses to glutamate [196]. In addition, phosphorylation of NR2B at Tyr1472 also activates reactive oxygen species (ROS) in a calcium-independent manner further supporting a detrimental role of Fyn in neuronal plasticity under neurotoxic stimuli [19,196].

Fyn phosphorylates AMPAR at Tyr876 (Y876) of GluR2 thus disrupting the association of GluR2 with glutamate receptor interacting proteins 1 and 2 (GRIP1/2), triggering endocytosis of GluR2 and ultimately leading to reduction of the surface expressed AMPARs [197].

Modulation of NMDA receptor controls cognitive function in prefrontal cortex and depends on dopamine D1 receptor [198]. The activation of dopamine D1 receptor triggers a rapid redistribution of NMDA receptors and increase NMDA expression at postsynaptic level [199].

Fyn phosphorylates the metabotropic glutamate receptor (mGluR) too [200,201]. Specifically, Fyn phosphorylates mGluR1a at Tyr937 (Y937) promoting its surface expression and controlling mGluR1a-dependent signaling transduction [200,201].

Fyn phosphorylation induces LTP in CA1 hippocampal areas [94,202]. Consistently, genetic ablation of *Fyn* results in LTP deficits [203] and gross structural changes in the dentate gyrus [163]. Interestingly, the role of Fyn appears to be restricted to specific developmental stages as it is not detectable in animals at less than 14 weeks [204].

Finally, Fyn activates protein tyrosine kinase 2 beta (Pyk2), encoded by the AD risk gene PTK2B, [205,206] and regulating synaptic plasticity. As the disease progresses, Aβ has been proposed to activate the Fyn phosphatase, striatal-enriched protein tyrosine phosphatase (STEP), eventually inactivating Fyn, which leads to the loss of synapses and dendritic spine collapse [207,208,209].

In mouse models, Fyn has been implicated as a downstream target of Amyloid β (Aβ) [209] (Figure 5). Accordingly, Aβ oligomers from the brains of AD patients activate Fyn after interacting with PrP(C) [177,210]. Aβ-Fyn interaction results in NR2B subunit phosphorylation, endocytosis of NMDARs and finally in synaptic deficiencies [182,183].

Notably, Fyn upregulation has been associated with increased APP Tyr682 residue phosphorylation in human AD neurons. Such hyperphosphorylation has been described to precede APP amyloidogenic processing, amyloid β accumulation, and neurodegenerative events [6,211] (Figure 5).

Fyn modulates cytoskeletal dynamics by phosphorylating and consequently inducing delocalization of proteins involved in cytoskeletal organization, such as Tau [212]. On the other side, Tau protein is hyperphosphorylated and abnormally folded in AD causing the lack in Tau ability to bind and stabilize microtubules in the axon. This loss of Tau function confers increasing aggregation properties triggering the formation of Tau tangles in AD [174,213] (Figure 5).

Although largely investigated, the role of Fyn in AD onset and/or progression is only partially understood. A large body of evidence has underlined the critical role of Fyn in balancing Tyr phosphorylation content of numerous neuronal proteins, including Tau and APP [6,214]. Consistently, the increased tyrosine phosphorylation of target proteins can be blocked by the addition of TK inhibitors suggesting that Fyn hyperactivity might be pharmacologically targeted to delay degenerative processes in AD [6,174,176,215,216].

Haass et al., recently described Fyn-APP/Aβ-Tau as a toxic triad. [217]. This toxicity appears to be related to the altered Tau redistribution in AD neurons, resulting in an enhanced Fyn-mediated phosphorylation of Tau either directly at the level of Tyr18 or indirectly at the level of Ser/Thre residues through GSK-3β activation [214,217,218,219].

Fyn expression in the brain is influenced by AD status and genetic content. An upregulation of Fyn and Tau proteins have been reported in a subset of neurons from AD tissues in the initial stages of neurodegenerative processes [214,218].

Fyn distribution and levels are altered in AD brains [220,221], and the genetic ablation of Fyn counteracts Aβ oligomers toxicity in hippocampal slices [222], suggesting that Aβ may derange synaptic functions through the aberrant activation of Fyn-related pathways [13,223]. Accordingly, an increased Fyn activity sensitizes neurons to Aβ-induced neuronal toxicity and exacerbates the Aβ-related increase in neuronal activity [13].

Interestingly, compensatory mechanisms that limit Fyn hyperactivity have been also reported in mice model of AD, overexpressing hAPP, consisting in a significant increase in phosphatase STEP levels, which dephosphorylates and inactivates Fyn [13] and consequently, reduces NMDA receptor phosphorylation and internalization and prevent synaptic dysfunctions [13,182,221].

miR-106b whose expression is reduced in AD tissue, directly binds to the 3′UTR of Fyn mRNA, and consequently deregulates Fyn mRNA expression resulting in decreased Fyn mRNA levels mostly in the frontal cortex. This finally results in a decreased Tau phosphorylation at Tyr18 [224].

Two SNPs located in the FYN gene, rs7768046, and rs1621289 (the first tagging the long isoform promoter region and the second located within the 3′ UTR region of FYN) appear to be associated with increased total Tau (t-Tau) levels in AD cerebrospinal fluid (CSF) [225].

Fyn phosphorylates also α-synuclein, a presynaptic protein that has been found hyperphosphorylated in neurons of Parkinson’s disease (PD) and AD patients [226]. Consistently, Fyn-mediated α-synuclein phosphorylation at Tyr125 was inhibited by the TyrKI inhibitor, PP2 [226]. In addition, studies performed in Fyn KO mice have prospected the possibility to use Fyn TK inhibitors to prevent or recover the altered dopamine-dependent trafficking of striatal NMDAR observed in PD neurons [227].

Of note, several reports have underlined the role of Fyn in several other pathological conditions of CNS, including neural trauma, prion disease, seizure by potentiating excitatory activity in the neurons through NMDARs and promoting hyperexcitation and neurotoxicity in epileptic brains [14,228].

## 4. Conclusions

In summary, there is mounting and compelling evidence implicating Fyn as a key factor in modulating physiologic neuronal pathways and development. These observations raise the questions of whether and which alterations in these molecular pathways lead to neuronal deficiencies and dysfunctions and whether targeting Fyn can have disease-modifying effects in these conditions. These questions seem to find an answer in prior work from several groups including some of our studies where it has been suggested that the pharmacologic targeting of Fyn may be therapeutically efficacious in dysfunctional neurons in which Fyn activity is impaired.

However, despite the encouraging evidence that targeting known and emerging elements of Fyn signaling cascades may become a promising therapeutic strategy in neuronal dysfunctions, Fyn still remains a challenging target, with broad expression throughout the body and significant homology with other members of the Src family kinases, likely leading to unintended off-target effects.

Overall, this review pinpoints the importance of Fyn in a multitude of neuronal function and dysfunction opening up the necessity of the development of innovative and selective therapeutic strategies to control Fyn activity and prevent brain diseases.

## Figures and Tables

**Figure 1 ijms-21-04444-f001:**
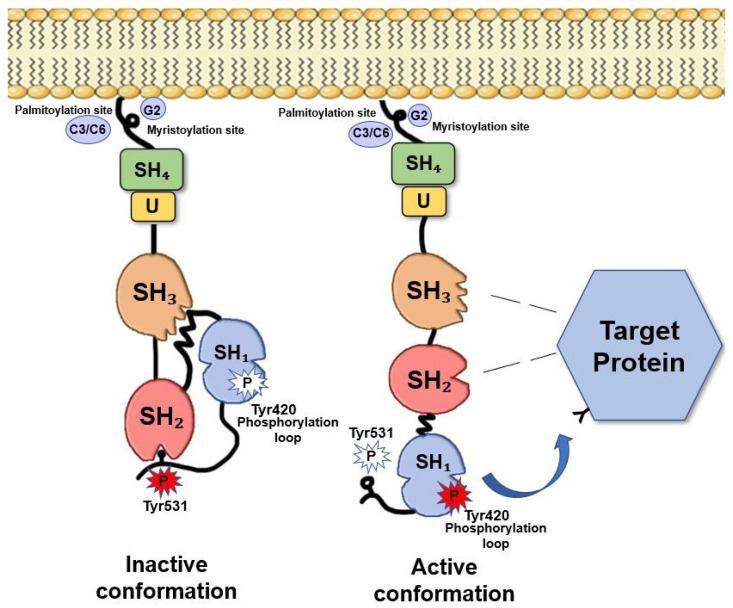
Fyn structure. The SH4 domain, is a membrane-targeting domain containing signals for the appropriate subcellular localization and membrane attachment of Fyn [21]. This domain consists of an extreme *N*-terminal ‘Met-Gly-Cys’ motif followed by a polar region [9,22]. The *N*-terminal Gly2 seems to be an absolute requirement for myristylation. Cys3 and Cys6, are both palmitoylated via a thioester linkage [23], thus, allowing Fyn anchors to the plasma membrane [24,25,26]. Between the SH4 and the SH3 motifs, there is a short region called “the unique region” that it is likely to be required for the subcellular localization of Fyn [27]. The SH3 motif constitutes a small domain of 50 amino acids containing a consensus sequence XPXXPPPXXP [28] that allows the binding to amino acid sequences rich in proline residues [29]. The SH2 domain consists of about 100 amino acids and appears to facilitate the binding to phosphotyrosine residues and to hydrophobic sequences within the cytoplasmic tails of growth factor receptors (i.e., PDGF-R, CSF-1 R) [30,31]. The catalytic domain SH1 is responsible of TK specific activity. This domain is highly conserved among Src family members [32]. The kinase domain, that catalyses the transfer of the terminal ATP phosphate group to a tyrosine residue of a target protein, shows a typical bilobed structure consisting in a small *N*-terminal lobe, involved in the binding with ATP, and a larger C-terminal lobe, where an activation loop (A-loop) is present. The conserved Tyr420, crucial for Src activity, is included in this domain [33]. However, the major site of phosphorylation and regulation of Fyn TK activity is the short negative regulatory tail where Tyr531 is located [32]. Adapted from Krämer-Albers et al. (2011) [34].

**Figure 2 ijms-21-04444-f002:**
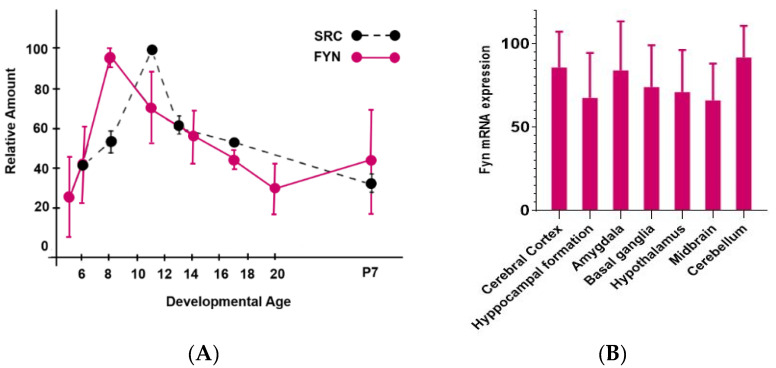
Fyn expression in human brain. (**A**) Fyn mRNA expression during development (6–20 refers to embryonic days; P7 = 7 days post-hatch). Adapted from Bixby et al. (1993) [58]. (**B**) Fyn human expression in the different areas of adult brain (data extracted from ATLAS).

**Figure 3 ijms-21-04444-f003:**
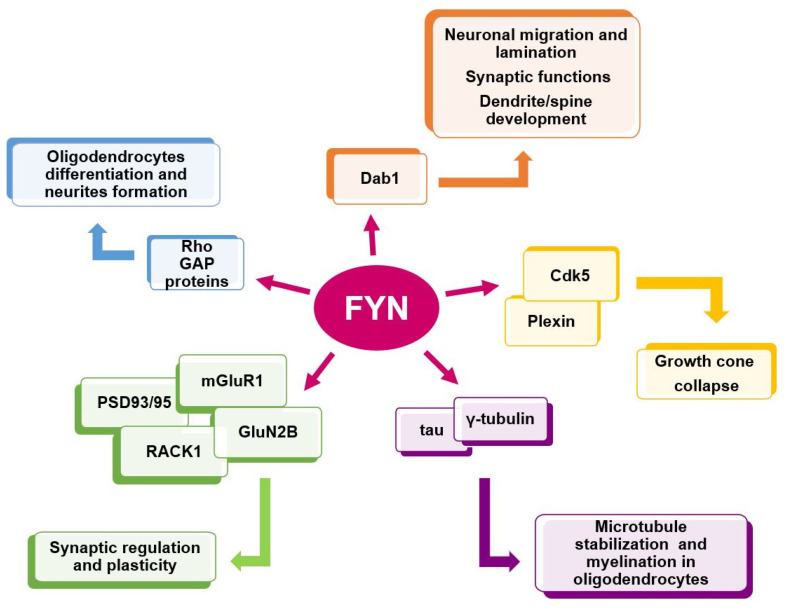
Fyn functions in the brain. Adapted from Schenone et al. (2011) [8].

**Figure 4 ijms-21-04444-f004:**
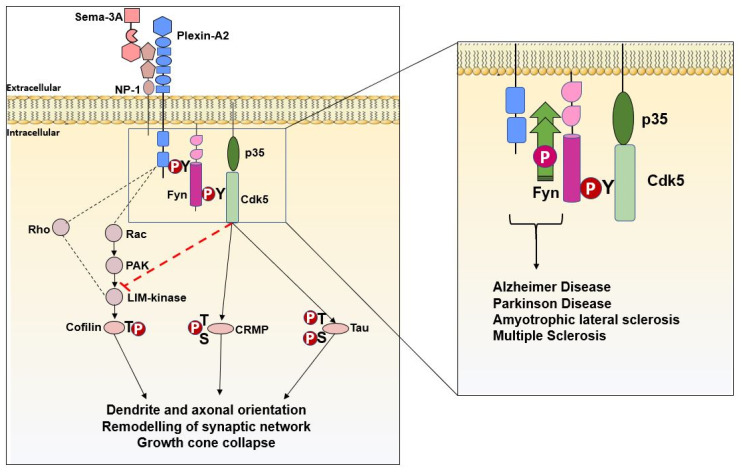
A proposed mechanism by which Fyn and Cdk5 control Sema3A and PlexinA2 downstream signaling cascade either in physiologic (left) or in pathologic (right) conditions. Fyn phosphorylates PlexinA2 and promotes PlexinA2 interaction to Sema3A. Fyn also phosphorylates cyclin-dependent kinase-5 (Cdk5) and in turn collapse response mediator protein (CRMP). The activation of Cdk5 facilitates the suppression of Rac-PAK signaling leading to the actin dynamic regulation. On the other hand, when phosphorylated, Cdk5 activates Tau resulting in destabilization of microtubules. Both these events culminate in growth cone collapse, remodeling of synaptic network and regulation of dendrites and axonal orientation (Adapted from Sasaki et al. 2002) [111].

**Figure 5 ijms-21-04444-f005:**
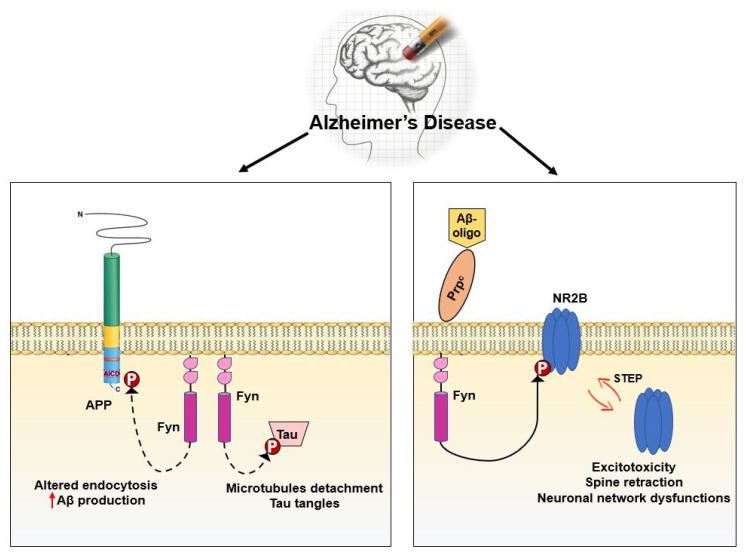
Fyn target proteins in AD neurons. Left: When overactivated, Fyn promotes phosphorylation either of APP at Tyr682 or Tau at Tyr18. These target protein hyperphosphorylations activate downstream pathways finally resulting in AD-related neuronal dysfunction and death. Right: A proposed mechanism by which Fyn, after interacting with the Aβ oligomers-Prion like protein complex, phosphorylates NR2B receptor and triggers excitotoxicity.

**Table 1 ijms-21-04444-t001:** PlexinA4 (PLEXA4), semaphorin 5A (SEMA5A) and Fyn-related kinase (FRK) gene association analysis in ASD patients. Adapted from Schafer et al. (2019) [97].

Symbol and Name Gene	Support in Autism	N° of Studies Reporting the Evidence	Supporting Evidence
**PLEXA4** (Plexin A4)	Functional	3	Copy number variations (CNVs) involving the PLXN-A4 gene were identified in two unrelated ASD case
**SEMA5A** (Semaphorin 5A)	Functional	15	Expression of the SEMA5A gene has been shown to be downregulated in some autistic individuals
**FRK** (Fyn-related kinase)	Genetic association	3	Genetic association has been found between the FRK gene and autism in two large cohorts (AGRE and ACC) of European ancestry and replicate in two other cohort (CAP and CART)

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
