# Peer review of "Fyn Tyrosine Kinase as Harmonizing Factor in Neuronal Functions and Dysfunctions"

_ijms, 2020, doi:10.3390/ijms21124444_

Round 1

Reviewer 1 Report

In this current review the authors summarized data suggesting that Fyn tyrosine kinase is a key factor involved in neuronal functions and dysfunctions.

This short review is clear and well organized. I have only mirror concerns:

  • In fig2b showing the Fyn expression level in human brain, the figure should include standard deviation in order to show the full distribution. These data could be found in several databases (ensembl, brain atlas…)
  • In Table 1, what does “Number of evidences” mean? There is no explanation within the text.
  • Between the sentence 196 and 224, there is no question of Fyn? This part also includes Table 1 and Figure 4 which do not mention Fyn. These table and figure should be updated to figure out the link with Fyn.
  • Sentence 342: Please add a reference describing PTK2B as AD risk gene

Author Response

We thank Reviewer 1 for the positive comments. All the minor concerns have been addressed in this new revised version.

As for his/her suggestions:

1. SD have been added to the Figure 2, panel a.

2. text in the Table 1 has been edited and a few explanations have been included in the legend.

3. the section regarding semaphorin and plexin has been shortened, Figure 4 has been replaced with another in which the role of Fyn has been better contestualized.

4. Reference describing PTK2B as AD risk gene has been added.

Reviewer 2 Report

This review manuscript broadly describes functions of Fyn in the brain, such as myelination, synaptic regulation, and so on. Since Fyn seems to play an important role in Alzheimer's disease, this review could help researchers understand.

However, I think the revision will improve this manuscript to be better.

Major:

  1. Recently, not only in vascular dementia but also in Alzheimer's disease, myelination impairment and oligodendrocyte dysfunction has been attracting attention (e.g. Allen M, et al. Alzheimers Dement Araque Caballero MA, et al. Brain 2018). Therefore, if there are any reports about an association between Fyn's myelination function and these cognitive disorders, it should be mentioned in section 3.1. Even if not yet known, it should be mentioned as it remains unknown.
  2. Page 10, Line 297: This paragraph describes the role of Fyn in synaptic disfunction in Alzheimer's disease. Therefore, this should be one of the most important part in this review. It should be described in more detail by adding more citations (e.g. Ittner LM, et al. Cell 2010) and adding a figure (such as Supplemental Figure S10 in Ref. 191: Um JW, et al. Nat Neurosci 2012) if possible.

Minor:

Page 6, Line 153: The abbreviation for MBP has already been explained on line 128.

Page 10, Line 300: Aβ is garbled.

Page 11, Line 348: Same as above.

Author Response

We appreciate the suggestions of the reviewer 2 and we revised the review accordingly.

In this revised version

1. we included some studies suggesting an association between dysfunctions in Fyn mediated myelination processes and neurodegenerative disorders

2. we extended the paragraphs related to the role played by Fyn in AD as well as in other neurodegenerative diseases.

All the minor points are now corrected.

Reviewer 3 Report

The manuscript entitled “Fyn tyrosine kinase as harmonizing factor in neuronal functions and dysfunctions”, is a hot topic with great scientific relevance. This paper is very descriptive, but to stand out from the other published reviews, in my opinion, it should be pointed out more study cases, mentioning the role of Fyn in neuronal functions and dysfunctions, emphasizing the title proposed for this work. Moreover, in this manuscript, there are many sentences completely transcribed from other publications, that should be reviewed. Some are listed below for consideration.  

  1. “S. Schenone, C. Brullo, F. Musumeci, M. Biava, F. Falchi, M. Botta. Fyn Kinase in Brain Diseases and Cancer: The Search for Inhibitors, Current Medicinal Chemistry, 2011”, such as in lines 27-29; 45-46, 58-62, 66-67 and 69-70.
  2. “James Reinecke, Steve Caplan. Endocytosis and the Src family of non-receptor tyrosine kinases, Biomolecular Concepts, 2014” – lines 37-38;
  3. “Cooper, J.A.. Src Family of Protein Tyrosine Kinases, Encyclopedia of Biological Chemistry, 2013” – lines 32-33;
  4. “Jeffrey A. Zahratka, Yvonne Shao, McKenzie Shaw, Kaitlin Todd et al. Regulatory region genetic variation is associated with FYN expression in Alzheimer's disease, Neurobiology of Aging, 2017”– lines 300-303 and 362-365;

I also recommend taking care of the following minor suggestions:

On page 2 line 51, change the word “detected” by “identified” and in line 52 “we will restrict our interest (…)” by “we focus on (…)”.

In the section “2. Fyn structure and activation mechanisms” pay attention to the capital letter of “Fyn” in lines 59, 60 and 63 and in line 67 the sentence “catalytic domain” is repeated, and thus should be removed.

On page 4, lines 87-88, this sentence is awkwardly positioned. Please revise accordingly.

The abbreviations “PDGF” referred in line 96 has not been mentioned previously in any part of the text.

The last paragraph on page 4 (lines 112-114) does not directly reflect what figure 2 means. It should be reformulated.

In page 10, line 284 what kind of animal models were used? A more detailed description would be expected.

Some references do not have the journal abbreviation and the respective DOI.

Author Response

We thank referee 3 for pointing on these relevant aspects.

In this revised version of the review, we included more evidence to reinforce the concept that Fyn can balance physiologic and pathologic brain processes in order to make neurons work properly.

We apologize for some inaccuracies that were marked in yellow in the previous version and that have been all edited in this revised version

Minor point notes have been amended

Round 2

Reviewer 2 Report

The authors fully responded to the comments.

Author Response

We thank reviewer 2 for his&her comments.

Reviewer 3 Report

The corrections were attended and therefore, in my opinion, the manuscript deserves to be published in International Journal of Molecular Sciences.

However, I suggest to reconfirm the numbering of references throughout the text (e.g. lines 42, 187, 370, 383) and pay particular attention to the references in lines 387-389, that should be referenced in numerical style.

Author Response

We thank reviewer 3 for his/her comments.

In this revised version, references have been reviised carefully